# Pathophysiology of Takotsubo Syndrome as A Bridge to Personalized Treatment

**DOI:** 10.3390/jpm11090879

**Published:** 2021-08-31

**Authors:** Monika Budnik, Radosław Piątkowski, Dorota Ochijewicz, Martyna Zaleska, Marcin Grabowski, Grzegorz Opolski

**Affiliations:** Department of Cardiology, Medical University of Warsaw, 02-097 Warszawa, Poland; radekp1@gmail.com (R.P.); dorota.ochijewicz@gmail.com (D.O.); zaleskamartyna@gmail.com (M.Z.); marcin.grabowski@wum.edu.pl (M.G.); grzegorz.opolski@wum.edu.pl (G.O.)

**Keywords:** tatotsubo syndrome, pathophysiology, stress’ heart-brain axis

## Abstract

Takotsubo syndrome (TTS) consists of transient dysfunction of the left and/or right ventricle in the absence of ruptured plaque; thrombus or vessel dissection. TTS may be divided into two categories. Primary TTS occurs when the cause of hospitalization is the symptoms resulting from damage to the myocardium usually preceded by emotional stress. Secondary TTS occurs in patients hospitalized for other medical; surgical; anesthetic; obstetric or psychiatric conditions who have activation of their sympathetic nervous system and catecholamines release- they develop TTS as a complication of their primary condition or its treatment. There are several hypotheses concerning the cause of the disease. They include a decrease in estrogen levels; microcirculation dysfunction; endothelial dysfunction and the hypothesis based on the importance of the brain-heart axis. More and more research concerns the importance of genetic factors in the development of the disease. To date; no effective treatment or prevention of recurrent TTS has been found. Only when the pathophysiology of the disease is fully known; then personalized treatment will be possible.

## 1. Introduction

Takotsubo syndrome (TTS) is a transient dysfunction of the left and/or right ventricle, without the presence of ruptured atherosclerotic plaque, thrombus, dissection of the artery or other condition, which may cause the aforementioned symptoms. The term “tako-tsubo” was first used by Sato et al in 1990 because of the similarities between LV morphologic features and the shape of a ceramic fishing pot used since ancient times in Japan for trapping octopi [1]. Diagnostic criteria are presented in Table 1. There are multiple clinical scenarios of the course of TTS, which can be classified as either primary—when the primary reason of the hospitalization is symptoms of the acute myocardium injury, mainly triggered with emotional stress, or secondary—the individual is already hospitalized for other medical conditions and as a result of the sudden activation of the sympathetic nervous system (SNS), a rise in catecholamines reversible contractile dysfunction occurs (Table 1).

The aim of the article is to review the current literature on typical clinical features and pathophysiology of TTS. Only the elucidation of the pathophysiology of the disease may be the basis for personalized treatment.

## 2. Clinical Findings

One of the characteristic features of TTS is its association with stress factors triggering the onset of the disease. According to the InterTAK classification, TTS may be classified into three groups based on the type of triggering factor: associated to emotional trigger, physical trigger or with no evident trigger (Table 2) [3]. One should notice that in many cases we see a co-occurrence of both emotional and physical triggers. In approximately 1/3 of patients the trigger cannot be identified [3]. Emotional factors are comprised of multiple, different traumatic experiences, such as mourning, interpersonal conflicts, fear, panic, anxiety, and disappointment. Natural disasters, such as earthquakes or floods, also predispose one to TTS development [4,5,6]. The aforementioned negative emotions that are the reason of TTS gave rise to the name ‘broken heart syndrome’ [7]. Emotional stress may lead to the overstimulation of the SNS and/or to a reduction of parasympathetic nervous system activity. This could result in life-threating arrhythmias, TTS, or even sudden cardiac death [8]. Although, emotions, which are direct triggers of TTS, do not need to be negative. In literature we can find a range of case reports where TTS was associated to strong positive experiences. Sometimes these cases were described as ‘happy heart syndrome’ [8]. Positive emotions modulate the autonomic response, including heart rate and blood pressure, in a similar manner as the negative ones [9].

The most common sign of TTS is chest pain, which is observed in approximately 75% of patients. Almost half of the individuals report dyspnea. Additionally, one may observe dizziness (ca. 25%) or syncope (ca. 10%) [10]. It must be emphasized that the first, and the only manifestation of TTS could be sudden cardiac death.

Electrocardiographic (ECG) changes usually include ST-segment elevation, T-wave inversion, and QT-interval prolongation. Less often one may observe a left bundle branch block, atrial fibrillation, or ST-segment depression. The absence of any deviations in the ECG does not exclude a TTS diagnosis.

In the most typical scenario of TTS, ECG dynamics include:

Phase 1—ST-segment elevation;

Phase 2—T-wave inversion (new onset of negative T-waves)—usually within 1–3 days;

Phase 3—T-waves reversion—usually within 2–6 days;

Phase 4—Development of deeply inverted T-waves, with QT-interval prolongation—usually within 2 months, up to recovery [11].

ST-segment elevation is mainly observed in leads V3–V6 [12]. Although, in comparison to patients with ST-segment elevation myocardial infarction (STEMI), the amplitude of the ST-segment elevation is usually lower and is observed in fewer leads [13,14]. An index proposed by the Japanese team (amplitude of the ST-segment elevation in leads V3–V6 in the ratio to amplitude of the ST-segment elevation in leads V1–V3) is higher in patients diagnosed with TTS and shows 100% specificity and 91% sensitivity in detecting TTS [12]. The least characteristic for TTS is ST-segment elevation in V1, which is definitely more common in patients with anterior wall STEMI [15,16,17].

ST-segment depressions are usually observed within 24–72 h after presentation of the symptoms. In the European TTS Registry, ST-segment depression was observed more rarely in patients diagnosed with TTS, when compared to individuals with acute coronary syndrome (ACS) [18], and more frequently in male patients [19].

The development of negative T-waves is usually accompanied by QT-interval prolongation [20]. Significant QT-interval prolongation carries a risk of torsade de pointes [21]. In literature we can also find descriptions of sustained ventricular tachycardias in the course of TTS [22].

Almost every patient diagnosed with TTS has elevated cardiac enzymes, in which the concentration is unproportionally low in relation to the amount of the myocardium area affected by contractile dysfunction. Typically, we observe high ratio of NT-proBNP concentration to troponin I (TnI) concentration [23].

Echocardiography is the basic imaging examination used for TTS diagnosis. We can differentiate four major anatomical variants of TTS: apical, mid-ventricular, reverse and focal. The most common type is apical TTS, which is diagnosed in almost 80% of patients (Figure 1 and Figure 2). In every variant of TTS, segmental contractile dysfunction usually exceeds the area supplied by one coronary artery.

Right ventricle dysfunction is observed in approximately 30% of patients with TTS [24]. It should be emphasized that due to hyperkinesis of the right ventricle inflow tract, standard parameters used for right ventricle function assessment, such as tricuspid annular plane systolic excursion (TAPSE) and Doppler tissue imaging-derived tricuspid annular systolic velocity (TDIs’), may not be useful. In patients with biventricular TTS right ventricle contractile disturbances mirror these observed in the left ventricle (akinesia of right ventricle outflow tract, with hyperkinesia of inflow tract) (Figure 3). The aforementioned disturbances are contrary to the McConnell sign (hyperkinesia of right ventricle outflow tract, with akinesia of the inflow tract), which is stated in patients with pulmonary embolism, so some researchers use the term ‘reverse McConnell sign’ [25].

Until now, despite multiple studies, the exact cause of TTS remains unknown. There are a few hypotheses, which aim to explain the underlying grounds of the disease.

## 3. Pathophysiology

### 3.1. Decrease of Estrogen Level

It is characteristic that approximately 90% of patients are elderly, post-menopausal women [26]. Multiple experimental, as well as clinical studies, propound the function of female sex hormones as crucial in TTS pathophysiology. It seems that a decrease in estrogens levesl is critical. Currently, in literature we can find few clinical studies evaluating the concentration of sex hormones in patients with TTS. Additionally, their results remain inconclusive.

Brenner et al. proved that estrogen concentrations at admission were significantly higher in post-menopausal women diagnosed with TTS, when compared to patients with myocardial infarction (MI) and the control group (subjects without lesions in coronary arteries). They did not observe significant differences in progesterone, luteinizing hormone, or gametokinetic hormone concentrations between groups. Furthermore, in a 6 year follow-up among patients diagnosed with TTS, they noticed significantly lower estradiol concentrations when compared to concentrations at admission, and lower progesterone levels in comparison to groups with MI or the control group [27]. On the other hand, Möller et al. did not show significant differences in estrone, estradiol, testosterone or androstenedione concentrations between patients with TTS and individuals with MI [28]. Pizzino et al. observed that specific polymorphisms of estrogen receptors are associated to TTS diagnosis in post-menopausal Caucasian women [29]. Studies showed that hormone replacement therapy in post-menopausal women improves the endothelium-derived blood vessels’ expansion and increases myocardial perfusion [30]. On top of that both, clinical and experimental studies suggest significant influence of estrogens’ deficit on elevated SNS activity, which seems to play a key role in TTS pathogenesis.

### 3.2. Microcirculation Disturbances

Another hypothesis includes microcirculation disturbances. In one of the studies, the authors analyzed data from 18 patients in an acute phase of apical TTS (1–17 days). Their results showed a lack of perfusion disturbances in 10 individuals, and only mild dysfunction in 8. Perfusion disturbances were limited to apex and apical segments of the left ventricle [31].

Furthermore, they evaluated the Index of Microcirculation Resistance (IMR), and IMR 90 U indicated microcirculation dysfunction in patients with TTS (the normal range of IMR is <20–30 U). Additionally, the authors assessed the TIMI frame count (TFC; number of frames needed to move contrast from the ostium to the distal part of the vessel). Based on this study they proved that there are microcirculation disturbances in TTS patients when compared to a healthy population. Moreover, they suggest that a TFC > 20 differentiates TTS patients from healthy individuals. The dysfunction of microcirculation had diffused character in both TTS, as well as in microvascular angina, but it was more pronounced in microvascular angina. Unfortunately, it remains unknown whether microcirculatory dysfunction in TTS is the cause, or rather the consequence, of the disease [32].

### 3.3. Endothelium Dysfunction

In literature we can find only a few studies suggesting that endothelium dysfunction, expressed as an imbalance between vasoconstricting and vasodilating factors, may play an important role in TTS pathogenesis [33]. In the course of a TTS patient’s hospitalization researchers showed a significant increase of flow-mediated vasodilatation (FMD). The negative correlation between the FMD value and TnI concentrations at admission and the length of the hospitalization was also described [34]. A sudden decrease in FMD values at admission may probably be associated to a high concentration of circulating catecholamines, which impair endothelium function, contract vascular smooth muscles, and decrease their sensitivity to vasodilatory mediators, whereas an increase of FMD values at discharge may be caused by a gradual decrease of catecholamines concentration in the blood.

## 4. Low-Grade Inflammatory Process

A succedent theory involves low-grade inflammation. In literature we can find data suggesting that TTS may be associated to a classic inflammatory reaction, based on the results of experimental studies. Wilson et al. found that after isoprenaline administration in rats there was a significant increase in the number of neutrophiles, and subsequently of macrophages. Additionally, they found the number of M1 cells (so proinflammatory tissue destructive macrophages) increased over the time of observation after isoprenaline administration, with the peak at day 4. Whereas the population of M2 cells (anti-inflammatory, tissue reparative/profibrotic macrophages) presented a different pattern. Their numbers increased within the first day, then remained constant until day 6, and afterwards decreased since day 7. The authors also found that the percentage of M2 macrophages correlates with ejection fraction, which leads to the conclusion that the higher number of M2 cells improved muscle recovery. They found similar inflammatory changes in human patients diagnosed with TTS in histological postmortem examination, though they examined only 2 patients [35].

### 4.1. Brain-Heart Axis

The most probable hypothesis is based on the function of the brain-heart axis. Acute emotional stress cause brain activation, which results in an increase of cortisol, epinephrine, and norepinephrine bioavailability [36]. One of the proposed hypotheses assumes that a high level of the catecholamines induce stunning of the myocardium [37]. Decreased contractility had been observed during supraphysiological levels of epinephrine as the β2-adrenoreceptor (ADRB2) signaling pathway switches from the Gs to the Gi activation. This pathway blocks cAMP production, hence reversing the effects of Gs stimulation, as well as targeting pathways that promote cell survival [38]. The variability in the apical-basal density of sympathetic nerve endings, cardiomyocyte membrane organization, receptor densities and types (β1- and β2-adrenoreceptors) are likely the cause of differential responses of heart regions [39].

Moreover, the stimulation of β-adrenoreceptors has an influence on cardiac lipid metabolism—it accelerates lipolysis and increases lipid uptake and oxidation. Overstimulation of the aforementioned receptors associated to supraphysiological levels of catecholamines may disturb lipid metabolism and lipid homeostasis in the heart. Shao et al. proved that catecholamines-induced lipotoxicity may play an important role in TTS pathogenesis. They found that after isoprenaline administration there was a severe lipid accumulation in the myocardium in mice which normalized at recovery. This may be caused by a significant decrease in the expression of the ApoB gene (gene of one of the myocardial lipoproteins involved in lipid export). These findings were confirmed in patients diagnosed with TTS [40].

Suzuki et al. measured the regional blood flow in the brains of a small group of patients in the acute phase of TTS. They showed a significant increase of blood flow in the hippocampus, brain trunk and basal ganglia, simultaneously with a decreased blood flow in the prefrontal area [41].

Based on the magnetic resonance imaging (MRI) of the head in patients with a TTS history and the control group consisting of age-matched, healthy individuals, Templin et al. evaluated differences in the brain regions responsible for SNS activity, regions responsible for emotions, and limbic and autonomic systems integration. Analysis of the structural brain connectivity confirmed a reduction of connections in the limbic system [42]. Furthermore, the analysis of connections specific for autonomic systems showed their reduction in patients diagnosed with TTS in the region of the left amygdala, both hippocampi, left parahippocampal gyrus, left superior temporal pole, and right putamen. The voxel-wise morphometry showed a reduced gray-matter volume in TTS in the left amygdala, right amygdala and at the right amygdala/hippocampus border [43]. Anatomical differences in the limbic system structure may be crucial for less effective control of the emotions, cognitive functions, and autonomic nervous system.

The aforementioned theory could be directly confirmed with the fact that cingulate gyrus is one of the structures, whose dysfunction may be responsible for the development of depression and mood disorders, whose cooccurrence is common among patients with TTS as well [44]. Takotsubo syndrome occurrence could be secondary to conditions with co-existing increases of SNS activity (e.g., pheochromocytoma, paraganglioma, subarachnoid hemorrhage, stroke, epileptic seizure, migraine, and postural orthostatic tachycardia syndrome). Takotsubo syndrome could develop under other circumstances associated with the increase in sympathetic activity, such as drug or alcohol withdrawal syndrome, during electroconvulsive therapy, or after cocaine abuse. It is quite interesting that diabetes prevalence is significantly lower in the group of patients diagnosed with TTS, when compared to the general population. It was suggested that peripheral diabetic autonomic neuropathy may reduce the number of TTS cases throughout, decreasing the effect of adrenergic heart overstimulation.

In noninvasive tests of SNS activity, the authors proved that during the Valsalva manoeuvre, static handgrip exercises or tilt tests, the change in pulse pressure (difference between systolic and diastolic pressure values) was lower in patients with TTS, when compared to the control group (consisting of healthy, age-matched women). Pulse pressure depends on stoke volume and peripheral resistance. The aforementioned study’s results could be considered important, because thanks to these noninvasive tests, we may indirectly assess autonomic system function, so by extension, recreate common real-life situations. An impaired pressor response may be caused by heart weakness in pumping function caused by an inadequate autonomic system reaction to stress [45].

### 4.2. Genetic Factors

A potential genetic predisposition to TTS has been suggested by the presence of familial cases, recurrence of TTS in 5–10% of cases and the fact that very few people develop TTS despite experiencing stressful events throughout their lives [3,46,47,48,49]. However, genetic studies have provided conflicting results concerning the potential association between TTS and gene polymorphisms. The previous studies have reported in TTS patients single nucleotide polymorphisms in genes associated with sympathetic regulation, such as β1-adrenoreceptor (ADRB1), β2-adrenoreceptor (ADRB2), a2C- adrenoreceptor (ADRA2C), G-protein-coupled receptor kinase 5 (GRK5) and antiapoptotic protein BcI-associated athanogene 3 (BAG3) [50,51,52,53]. Borchert et al. showed enhanced β -adrenergic signaling and higher sensitivity to catecholamine-induced toxicity in the in vitro-induced pluripotent stem cell (iPSC) model of TTS [54]. Though several studies found a lack of genetic susceptibility in TTS patients compared with the controls [55,56,57,58]. Given the weak evidence further studies with larger sample sizes are warranted to reveal the potential genetic alterations contributing to TTS.

Singh et al. published metanalysis, where they found that TTS recurrence oscillate between 1.2% in the first 6 months after primary diagnosis, up to 5% after 6 years of follow-up. The mean age of patients when they were diagnosed with a second episode of TTS was 65.5 years. Recurrence appears in both, male and female patients, at every age (in childhood too), and may be caused by the same or different triggers and involve various anatomical variants of contractile disturbances [59,60]. They could be diagnosed within 3 weeks or even after 10 years since the first episode [59,60]. In literature we can find a case report of a patient, who developed TTS fivefold [61,62]. Interestingly, consecutive episodes were triggered by different types of factors (thrice it was sudden, powerful emotional stress, once long-lasting psychological stress and once physical activity). Hitherto causal treatment remains unknown. A Full understanding of the culprit of the disease could be an introduction to personalized treatment in TTS patients.

## 5. Conclusions

There are many hypotheses trying to explain the cause of takotsubo syndrome. At present, none has been fully proven. The most likely cause of TTS appears to be the impairment of the heart-brain axis. Perhaps in future it will be possible to block pharmacologically the effect of adrenergic heart overstimulation.

## Figures and Tables

**Figure 1 jpm-11-00879-f001:**
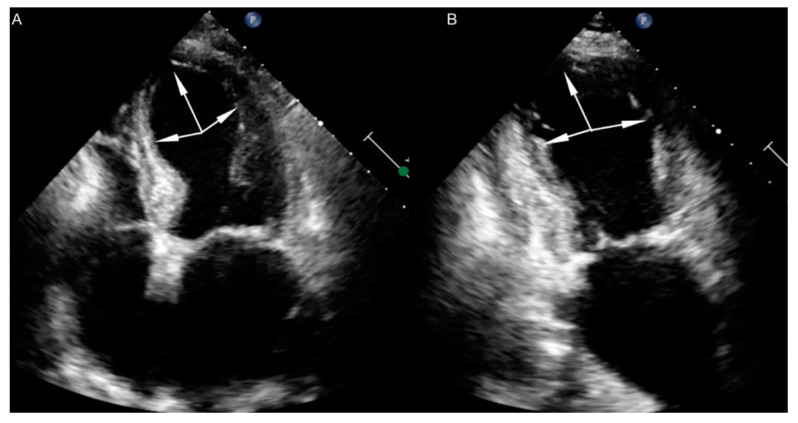
Echocardiography in a patient with typical (apical) form of TTS. (**A**) apical four chamber view. (**B**) apical two chamber view. The arrows show the region of contractility disturbance.

**Figure 2 jpm-11-00879-f002:**
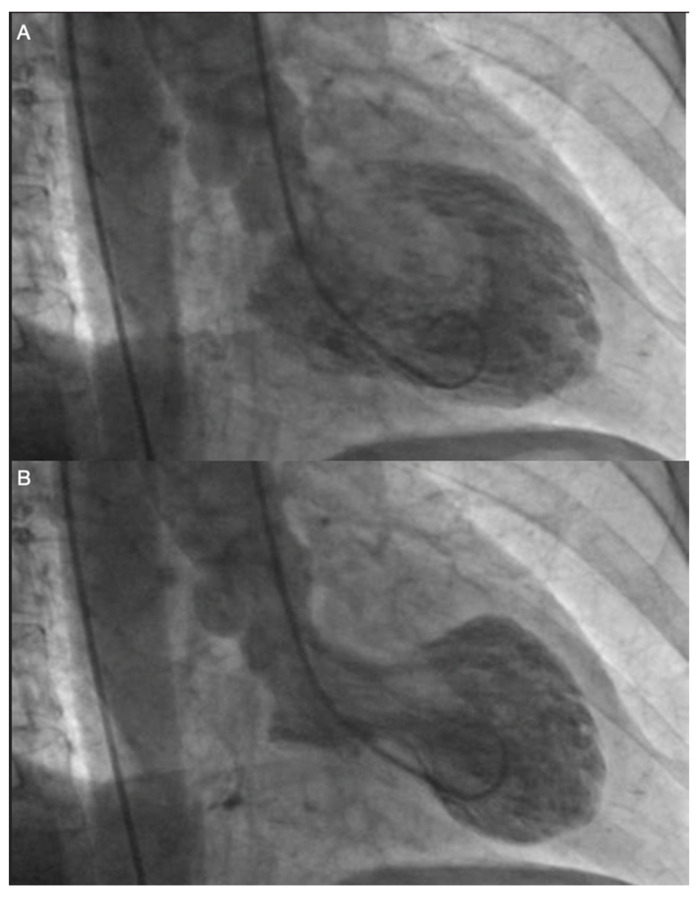
Angiography in a patient with typical (apical) form of TTS. (**A**) left ventricle in diastole. (**B**) left ventricle in systole.

**Figure 3 jpm-11-00879-f003:**
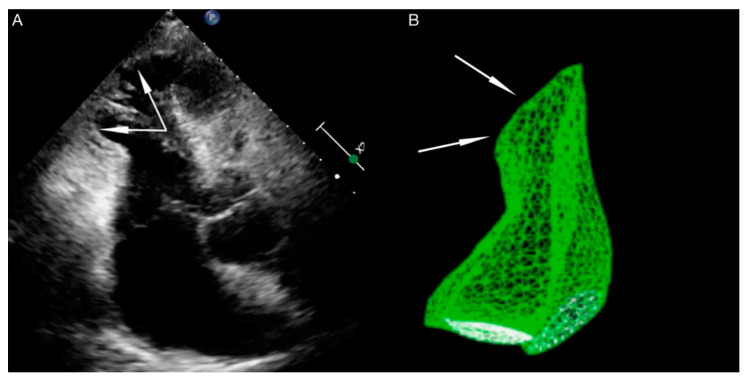
Echocardiography in a patient with right ventricle dysfunction (Authors’ own material). (**A**) 2D echocardiography, right ventricle view. (**B**) 3D echocardiography, right ventricle model. The arrows show the region of contractility disturbance.

**Table 1 jpm-11-00879-t001:** International Takotsubo Diagnostic Criteria [2].

International Takotsubo Diagnostic Criteria (InterTAK Diagnostic Criteria) ^a^
Patients show transientleft ventricular dysfunction (hypokinesia, akinesia, or dyskinesia) presenting as apical ballooning or midventricular, basal, or focal wall motion abnormalities. Right ventricular involvement can be present. Besides these regional wall motion patterns, transitions between all types can exist. The regional wall motion abnormality usually extends beyond a single epicardial vascular distribution; however, rare cases can exist where the regional wall motion abnormality is present in the subtended myocardial territory of a single coronary artery (focal TTS) ^b^; An emotional, physical, or combined trigger can precede the takotsubo syndrome event, but this is not obligatory; Neurologic disorders (e.g., subarachnoid haemorrhage, stroke/transient ischaemic attack, or seizures) as well as pheochromocytoma may serve as triggers for takotsubo syndrome; New ECG abnormalities are present (ST-segment elevation, ST-segment depression, T-wave inversion, and QTc prolongation); however, rare cases exist without any ECG changes; Levels of cardiac biomarkers (troponin and creatine kinase) are moderately elevated in most cases; significant elevation of brain natriuretic peptide is common; Significant coronary artery disease is not a contradiction in takotsubo syndrome; Patients have no evidence of infectious myocarditis ^b^; Postmenopausal women are predominantly affected.

^a^: Wall motion abnormalities may remain for a prolonged period of time or documentation of recovery may not be possible. For example, death before evidence of recovery is captured. ^b^: Cardiac magnetic resonance imaging is recommended to exclude infectious myocarditis and diagnosis confirmation of takotsubo syndrome.

**Table 2 jpm-11-00879-t002:** InterTAK Classification [1].

InterTAK Classification
Class I: Takotsubo syndrome related to emotional stress Class II: Takotsubo syndrome related to physical stress Class IIa: Takotsubo syndrome secondary to physical activities, medical conditions, or procedures Class IIb: Takotsubo syndrome secondary to neurologic disorders Class III: Takotsubo syndrome without an identifiable triggering factor

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
