# Peer review of "Pathophysiology of Takotsubo Syndrome as A Bridge to Personalized Treatment"

_jpm, 2021, doi:10.3390/jpm11090879_

Round 1
Reviewer 1 Report
The article "New mechanisms of takotsubo syndrome as a bridge to personalized treatment" by Monika Budniket al, is a review of the state of the art. However, the title is misleading since no reference is made to new mechanisms, beyond those already known, and no reference is made to personalised treatments for this pathology. In this sense, the paper raises more expectations than it fulfils. The review is adequate to current knowledge but the authors should propose a less ambitious and misleading title.
Author Response
I changed the title by changing the word "new mechanisms of takotsubo syndrome" to: "Pathophysiology of takotsubo syndrome as a bridge to personalized treatment"
Reviewer 2 Report
The authors aimed in this paper aimed to highlight the pathophysiolgical mechanisms of takotsubo syndrome.
Several flaws can be detected in this paper. Several major points are missing too.
1) first, there is no clear structure for this review. There is no well defined Introduction part where the aims of the review in details should be described. Here the introduction looks like a copy of the abstract
2) This is a review with no conclusion, the authors passed directly from a paragraph where they discussed genetic factors to the end of manuscript. A clear conclusion with future suggestions regarding pathophysiology/implications for clinical management in TTS as well as summary of ongoing studies are missing
3) There should be a scientifically clear description for the TTS starting from the first description and hypothesis done by Sato et al 1990. Please refer to Sato et al 1990. Takotsubo-type cardiomyopathy due to multivessel spasm. Clinical Aspect of Myocardial Injury: From Ischemia to Heart Failure (1990), pp. 56-64
4) Recent important work discussing abnormalities of central autonomic nervous system in TTS is missing (T. Hiestand et al. 2018 Takotsubo syndrome associated with structural brain alterations of the limbic system)
5) Today there is a lot of discussion about the myocardiale responses to high epinephrine Levels and activation of β2AR-Gi signaling as well other mechanisms generating apical dysfunction in TTS pathphysiology. This important point should be more and clearly described in the review.
Check:
Land, et al Computational modeling of Takotsubo cardiomyopathy: effect of spatially varying beta-adrenergic stimulation in the rat left ventricle Am J Physiol Heart Circ Physiol.
Wright et al. 2018 Cardiomyocyte membrane structure and cAMP compartmentation produce anatomical variation in beta2AR-cAMP responsiveness in murine hearts Cell Rep, 23 (2018), pp. 459-469
6) Advances are also today made highlighting an abnormal myocardial metabolism in TTS. The focus was on the shutdown of mitochondrial metabolism in the hypokinetic or akinetic apical segments that might be a form of catecholamine or combined catecholamine- and ischemia-induced acute metabolic stunning. Such point is important and was not described in the review.
Check: Shao et al. A mouse model reveals an important role for catecholamine-induced lipotoxicity in the pathogenesis of stress-induced cardiomyopathy
Eur J Heart Fail, 15 (2013), pp. 9-22
7) The role of low grade inflammation is also a promising issue in the pathophysiology of TTS. This point is also missing in this review.
Check for example the work of Wilson et al 2018.Characterization of the myocardial inflammatory response in acute stress-induced (Takotsubo) cardiomyopathy J Am Coll Cardiol Basic Trans Sci, 3 (2018), pp. 766-778
8) What is the source/reference of the mentioned figures?
There is no enough illustrating figures /tables in this paper. In addition angiographic images are somehow a must once talking about TTS.
Author Response
The authors aimed in this paper aimed to highlight the pathophysiolgical mechanisms of takotsubo syndrome.
Several flaws can be detected in this paper. Several major points are missing too.
- first, there is no clear structure for this review. There is no well defined Introduction part where the aims of the review in details should be described. Here the introduction looks like a copy of the abstract
We corrected the structure of the article.
2) This is a review with no conclusion, the authors passed directly from a paragraph where they discussed genetic factors to the end of manuscript. A clear conclusion with future suggestions regarding pathophysiology/implications for clinical management in TTS as well as summary of ongoing studies are missing.
We added conclusions.
3) There should be a scientifically clear description for the TTS starting from the first description and hypothesis done by Sato et al 1990. Please refer to Sato et al 1990. Takotsubo-type cardiomyopathy due to multivessel spasm. Clinical Aspect of Myocardial Injury: From Ischemia to Heart Failure (1990), pp. 56-64
We added this valuable research.
4) Recent important work discussing abnormalities of central autonomic nervous system in TTS is missing (T. Hiestand et al. 2018 Takotsubo syndrome associated with structural brain alterations of the limbic system)
We have discussed recent studies about abnormalities of central autonomic nervous system in TTS within the section ‘Brain-heart axis’ including aforementioned work
5) Today there is a lot of discussion about the myocardiale responses to high epinephrine Levels and activation of β2AR-Gi signaling as well other mechanisms generating apical dysfunction in TTS pathphysiology. This important point should be more and clearly described in the review.
Check:
Land, et al Computational modeling of Takotsubo cardiomyopathy: effect of spatially varying beta-adrenergic stimulation in the rat left ventricle Am J Physiol Heart Circ Physiol.
Wright et al. 2018 Cardiomyocyte membrane structure and cAMP compartmentation produce anatomical variation in beta2AR-cAMP responsiveness in murine hearts Cell Rep, 23 (2018), pp. 459-469
Thank you for this valuable comment. We have added proposed mechanisms related to high epinephrine levels and myocardial dysfunction in TTS in ‘Brain-heart axis’ section
6) Advances are also today made highlighting an abnormal myocardial metabolism in TTS. The focus was on the shutdown of mitochondrial metabolism in the hypokinetic or akinetic apical segments that might be a form of catecholamine or combined catecholamine- and ischemia-induced acute metabolic stunning. Such point is important and was not described in the review.
Check: Shao et al. A mouse model reveals an important role for catecholamine-induced lipotoxicity in the pathogenesis of stress-induced cardiomyopathy
Eur J Heart Fail, 15 (2013), pp. 9-22
Thank you for pointing this issue. We added a brief description of aforementioned mechanism in proper section of the manuscript (page 7).
7) The role of low grade inflammation is also a promising issue in the pathophysiology of TTS. This point is also missing in this review.
Check for example the work of Wilson et al 2018.Characterization of the myocardial inflammatory response in acute stress-induced (Takotsubo) cardiomyopathy J Am Coll Cardiol Basic Trans Sci, 3 (2018), pp. 766-778
Thank you for this comment. We added a separate paragraph discussing this issue.
8) What is the source/reference of the mentioned figures?
We added reference
There is no enough illustrating figures /tables in this paper. In addition angiographic images are somehow a must once talking about TTS.
We added angiography.
Round 2
Reviewer 2 Report
Legends to figures as well as references related to figures are still not clear or well organized. The conclusion is yet not suitable. Detailed suggestions considering the future therapeutic directions that are based on the pathophysiological mechanisms should be highlighted.
Author Response
Thank You for Your comments. We corrected figures legends as well as added possible therapeutic direction in "Conclusion" section.
